

# Depth to water table correction for initial carbon-14 activities in groundwater mean residence time estimation

Dylan J. Irvine[1,2], Cameron Wood[3], Ian Cartwright[4], Tanya Oliver[2]

[1]Research Institute for the Environment and Livelihoods, Charles Darwin University, Casuarina, 0810, Australia.
[2]National Centre for Groundwater Research and Training, and College of Science and Engineering, Flinders University, Bedford Park, 5042, Australia
[3]Department of Environment and Water, Adelaide, 5000, Australia
[4]School of Earth, Atmosphere and Environment, Monash University, Clayton, 3800, Australia

*Correspondence to*: Dylan J. Irvine (dylan.irvine@cdu.edu.au)

**Abstract.** Carbon-14 ($^{14}$C) is routinely used to determine mean residence times (MRTs) of groundwater. $^{14}$C-based MRT calculations typically assume that the unsaturated zone is in equilibrium with the atmosphere, controlling the input $^{14}$C activity. However, multiple studies have shown that unsaturated zone $^{14}$C activities are lower than atmospheric values. Despite the availability of unsaturated zone $^{14}$C data, no attempt has been made to generalise initial $^{14}$C activities with depth to the water table. We utilise measurements of unsaturated zone $^{14}$C

activities from 13 studies to produce a $^{14}$C-depth relationship to estimate initial $^{14}$C activities. The technique only requires the depth to the water table at the time of sampling, or an estimate of depth to water in the recharge zone to determine the input $^{14}$C activity, making it straightforward to apply. Applying this new relationship to two Australian datasets (113 $^{14}$C measurements in groundwater) shows that MRT estimates were up to 9250 years younger when the $^{14}$C-depth correction was applied relative to conventional MRTs. These findings may have

important implications for groundwater samples that suggest the mixing of young and old waters and the determination of the relative proportions of young and waters, whereby the estimated fraction of older water may be much younger than previously assumed. Owing to the simplicity of the application of the technique, this approach can be easily incorporated into existing correction schemes to assess the sensitivity of $^{14}$C$_{uz}$ to MRTs derived from $^{14}$C data.

## 1 Introduction

Environmental tracers are widely used to estimate both groundwater residence times (e.g., Love et al., 1994; Plummer and Sprinkle, 2001; Cartwright and Morgenstern, 2012; Jurgens et al., 2012) and recharge rates (e.g., Leaney and Allison, 1986; Cartwright et al., 2007; Gillon et al., 2009; Wood et al., 2015) that are important to effectively manage groundwater resources. Groundwater tracers can be used to determine mean residence times

(MRTs) from tens to hundreds of years (e.g., tritium or CFCs), thousands to tens of thousands of years (e.g., carbon-14) to millions of years (e.g., helium-4 or chlorine-36). In particular, carbon-14 ($^{14}$C) is widely used as a groundwater tracer owing to the ubiquitous presence of dissolved inorganic carbon in groundwater. With a half-life of 5730 years, $^{14}$C can be used determine residence times on the order of 1000 to 30,000 years, which encompasses the range of residence times in many regional aquifers (Clark and Fritz, 1997). The use of accelerator

mass spectrometry since the 1990s has significantly reduced the volumes of water required and has allowed highly precise measurements of $^{14}$C activities (on the order of ±1%), further facilitating the use of $^{14}$C as a groundwater tracer (Cartwright et al., 2020).





Although atmospheric $^{14}$C activities have varied over time (e.g., Clark and Fritz, 1997; Cartwright et al., 2017),
most studies of regional groundwater assume that prior to the atmospheric bomb testing in the 1960s they were constant at 100 percent modern Carbon (pmC). This approach yields conventional radiocarbon ages in years Before Present (BP) where 1950 AD = 0 years BP (Clark and Fritz. 1997; Plummer and Glynn, 2013). The subsequent input of $^{14}$C-free C from calcite dissolution within the aquifers may lower $^{14}$C activities and several schemes based on statistical corrections, major ion geochemistry, stable and radioactive isotopes exist to correct
for this (Ingerson and Pearson, 1964; Vogel, 1967; Tamers, 1967; Mook, 1972; Fontes and Garnier, 1979; Clark and Fritz, 1997; Coetsiers and Walraevens, 2009; Han and Plummer, 2016; McCallum et al., 2018). Locally geogenic $CO_2$ and/or the oxidation of old organic matter may also lower $^{14}$C activities (e.g., Clark and Fritz, 1997; Cartwright et al., 2017, 2020).

Further complications in the use of $^{14}$C to estimate MRTs relate to the input function of $^{14}$C into the subsurface. $^{14}$C activities of $CO_2$ in the unsaturated zone are typically assumed to be in equilibrium with the atmosphere at the time of recharge (e.g., Mazor, 2004). However, $^{14}$C activities of $CO_2$ in the unsaturated zone ($^{14}C_{uz}$) are commonly far lower than those of the atmosphere (e.g., Reardon et al., 1980; Haas et al., 1983; Thorstenson et al., 1983; Bacon and Keller, 1998; Carmi et al., 2009; Wood et al., 2014). For example, Carmi et al. (2009) note that $^{14}C_{uz}$
activities were approximately 54% of the atmospheric values in a coastal aquifer in Israel. Similar low $^{14}C_{uz}$ activities that decrease with depth below the land surface have been recorded elsewhere (e.g., Yang et al., 1985; Walvoord et al., 2005; Wood et al., 2014). Despite the important role of the unsaturated zone in controlling input $^{14}$C activities, this issue is rarely discussed, and no attempt has been made to produce a generalised relationship that relates initial $^{14}$C activities to depth to the water table.


The aim of this study is to produce a general relationship between $^{14}C_{uz}$ activity and the depth below surface to facilitate the estimation of input $^{14}$C activities to estimate MRTs. We produce a relationship using $^{14}C_{uz}$ and sample depth data from 13 studies across North America, Europe, the Middle East, and Australia. We provide a demonstration of the newly presented relationship using datasets from the Limestone Coast and Ovens/ Goulburn-
Broken catchments in Australia to estimate MRTs. For the Ovens/ Goulburn-Broken catchments dataset, we also present tritium data to provide supporting information for $^{14}$C-based calculations of MRTs. This work provides a simple to use approach to determine input $^{14}C_{uz}$ activities for groundwater MRT calculations that requires no additional measurement of groundwater chemistry. This approach can be incorporated into existing correction schemes to assess sensitivity of $^{14}C_{uz}$ to MRTs derived from $^{14}$C data.

## 2 Methods

### 2.1 Unsaturated zone data collation

$^{14}C_{uz}$ activities were collated from the following sources: Kukler, (1969); Fritz et al., (1978); Reardon et al. (1980); Haas et al., (1983); Gillon et al., (2009); Thorstenson et al., (1983); Yang et al., (1985); Leaney and Allison, (1986), Bacon and Keller, (1998); Thorstenson et al., (1998); Walvoord et al., (2005); Carmi et al., (2009); Wood
et al., (2014). A total of 181 $^{14}C_{uz}$ activities were collated from 14 sites across North America, Europe, the Middle



East and Australia (Fig. 1a). $^{14}C_{uz}$ activities were typically presented in tabular format in the original studies. In cases where data were presented graphically, $^{14}C_{uz}$ activities were obtained by digitising plotted results.

{Approx. location of Fig. 1}


The $^{14}C_{uz}$ study sites are predominantly located in North America (nine out of the 14 sites considered here), with two study sites from France (Gillon et al., 2009), two sites from Australia (Leaney and Allison, 1986; Wood et al., 2014) and a site in Israel (Carmi et al., 2009). A summary of the collated datasets is provided in Table 1. The complete $^{14}C_{uz}$ dataset is available in Table S1 in the Supporting Information.


{Approx. location of Table 1}

**2.2 Saturated zone data collation**

$^{14}C$ activities of groundwater samples ($^{14}C_{gw}$) were collated from the Limestone Coast region in South Australia and the Ovens/ Goulburn-Broken catchments in Victoria (Fig. 1b). Samples of $^{14}C_{gw}$ for the Limestone Coast

region were collated from Love et al. (1994), Dogramaci (1998), Brown et al. (2001), van den Akker (2006), Wood (2011), SKM (2012), Turnadge et al. (2013) and SA Water (2020). $^{14}C_{gw}$ activities from the Ovens/ Goulburn-Broken catchments are from Cartwright et al. (2007) and Cartwright and Morgenstern (2012). In total, 51 samples were collated from the Limestone Coast and 62 samples from the Ovens/ Goulburn-Broken catchments (Fig. 1b). The Ovens/ Goulburn-Broken catchments dataset also includes $^{3}H$ activities, which are useful for

understanding mixing between old and young groundwater (e.g., Jasechko et al., 2017).

Depth to water (DTW, presented in metres below ground level, mbg) values to accompany the measured $^{14}C_{gw}$ activities were included in Cartwright et al. (2007) and Cartwright and Morgenstern (2012) for the Ovens/ Goulburn-Broken catchments. For the Limestone Coast, DTW values were determined as follows: If detailed time

series of DTW were available for the sampled well, the measurement recorded as close in time to the $^{14}C_{gw}$ sampling was used. Where this was not possible, the average DTW for the sample year was used in the sample well, or a nearby well. All $^{14}C_{gw}$ and DTW data for the groundwater samples from the Limestone Coast and Ovens/ Goulburn-Broken catchments are provided in Table S2 in the Supporting Information.

**2.3 Data analysis**

The unsaturated zone sample depth-$^{14}C_{uz}$ relationship was produced by fitting the $^{14}C_{uz}$ and sample depth data (Table S1) using the *curve_fit* function in the *scipy.optimize* library and the *nominal_values* function from the *uncertainties.unumpy* libraries in Python. This approach also was used to find the best fit to the data, as well as to produce upper and lower bounds on the best fit relationship.

Not all collated $^{14}C_{uz}$ data were used to produce the $^{14}C$-depth relationships, with three principal reasons for the exclusion of data: (1) $^{14}C_{uz}$ activities were influenced by the presence of high levels of organic material in the unsaturated zone; (2) $^{14}C_{uz}$ activities were influenced by deep rooted vegetation, or; (3) where modern atmospheric





gases may have influenced the measured $^{14}C_{uz}$ activities. Modern air can pollute samples either through the drilling process, or through the advection of air through the unsaturated zone (e.g., Thorstenson et al., 1998). A description

of whether or not data values were used in the fitting process, and explanations for the omission of data values is provided in Table S1.

The compiled $^{14}C_{uz}$ dataset (Table S1) includes the depth below the surface of the sample, the measured $^{14}$C, and the sample year. It is expected that the $^{14}C_{uz}$ activities will be elevated where dissolved inorganic carbon influenced

by the bomb peak activities has entered the subsurface. The actual input function of $^{14}$C will differ from atmospheric inputs, as the $^{14}$C is first cycled through vegetation, with significant delays between the uptake of $^{14}$C in trees and entering unsaturated zone. For example, Fritz et al. (1978) measured $^{14}C_{uz}$ values of 141 ±10 pmC in the unsaturated zone in 1975, after peak $^{14}$C activities on the order of 180 pmC in the atmosphere in the mid-1960s (Fig. S1, Hua et al., 2013). However, owing to the relatively small sample size, the data was included in the fitting

process independent of the year in which it was collected.

$^{14}$C-based MRTs (y) were determined from the groundwater $^{14}$C measurements using the simple radioactive decay equation (Clark and Fritz, 1997):

$$\mathrm{MRT} = -8267\ln\left(\frac{^{14}C_{gw}}{^{14}C_i q}\right), \tag{1}$$

where $^{14}C_{gw}$ is the measured $^{14}$C activity in groundwater (pmC), $^{14}C_i$ is the initial $^{14}$C activity of the recharging water (pmC) and $q$ is the proportion of dissolved inorganic carbon that originated from groundwater recharge. While this approach neglects the fact that water follows variable flow paths and undergoes dispersion within the

aquifer and also assumes a uniform atmospheric $^{14}$C activity, it serves to illustrate the effects of variable $^{14}C_{uz}$. It is also the approach that is used in the majority of $^{14}$C studies (e.g., Cartwright et al., 2020). Again, for simplicity, we initially do not account for the input of $^{14}$C-free C from the aquifer matrix (i.e. $q = 1$); the impacts of the addition of $^{14}$C-free C is discussed in Section 3.3.

MRTs were calculated from the measured $^{14}C_{gw}$ activities using Eq. 1 firstly assuming that $^{14}C_i = 100$ pmC to produce conventional MRT estimates. Secondly, $^{14}C_i$ values determined from the DTW at the time of sampling using the DTW-$^{14}$C relationship derived in Section 3.1 were used to calculate the MRTs. MRTs were also estimated using the upper and lower bounds of the fitted relationship, producing upper and lower bounds on the MRT estimates based on the DTW corrections.


The lower limit of MRTs that can be estimated from $^{14}$C analyses is on the order of 1000-2000 years (Clark and Fritz, 1997; Cartwright et al., 2020). Here, MRTs less than 1000 years are considered to be "young water". For visualisation purposes, MRTs of <500 years are presented as 500 years as precise $^{14}$C-based MRTs cannot be determined for these samples. We adopt the Jasechko et al. (2017) definition of "fossil water" as water with MRTs

that exceed 12,000 years, which corresponds to the beginning of the Holocene.





## 3. Results

### 3.1 Development of depth-$^{14}$C relationship

The 181 unsaturated zone $^{14}$C activities are shown in Fig. 2. Shallow $^{14}$C$_{uz}$ data (< 20 mbg), particularly the shallow

values with very low $^{14}$C activities (< 25 pmC) were excluded from the data fitting process as the original articles suggest that these samples were influenced by oxidation of 'old' (low $^{14}$C activity) organic matter in the unsaturated zone sediments (Fig. 2). Unsaturated zone studies with data that were excluded from the data fitting process due to the presence of organic matter included Keller and Bacon (1988), Haas et al. (1983), Carmi et al. (2009). Other $^{14}$C activities were influenced by deep rooted vegetation (e.g., Leaney and Allison, 1986), and were

also excluded from the data that were used in the fitting process. Finally, data from the Yucca Mountain sites were omitted from the fitting process. Thorstenson et al. (1998) proposed that advection of modern air through the unsaturated zone has produced elevated $^{14}$C$_{uz}$ activities at the site. Thus, the relationships shown in Fig. 2 are representative of sedimentary basins and should generally not be applied where high organic matter or deep-rooted vegetation is present, or where advection of modern air occurs in the unsaturated zone.


{Approx. location of Fig. 2}

The best fit relationship ($R^2$ = 0.64) presented in Fig. 2 (solid line) can be represented mathematically as:

$$^{14}C_i = 104.63e^{-0.01693z},\qquad\qquad(2)$$

where $z$ is the depth below ground (mbg). The value 104.63 in Eq. 2 represents the $^{14}$C$_i$ at $z = 0$, which is reasonably consistent with present atmospheric $^{14}$C activities (e.g., Hua et al., 2013). The upper and lower bounds on Fig. 2 (dashed lines) were based on ±1 standard deviation (σ). The upper and lower bounds relationships can be

represented mathematically as:

$$^{14}C_i = 133.422e^{-0.01666z}, \text{ and}\qquad\qquad(3)$$

$$^{14}C_i = 82.606e^{-0.01720z}.\qquad\qquad(4)$$


The selection of 1σ was used rather than the more commonly used 2σ (representing upper and lower 95% confidence intervals) as the use of 2σ produced unrealistically high $^{14}$C$_{uz}$ activities in the shallow zone (0-1 m).

### 3.2 Application of depth-$^{14}$C relationship

Conventional MRTs determined assuming $^{14}$C$_i$ = 100 pmC (Eq. 1) and those using the DTW-correction (Eq. 2)

for the Limestone Coast and Ovens/Goulburn-Broken catchments are presented in Fig. 3.

{Approx. location of Fig. 3}



The application of Eq. 2 to determine $^{14}C_i$ values generally produces younger MRTs than the conventional MRTs (Fig. 3a, 3b). Exceptions to this observation occurred in cases where the DTW-corrected $^{14}C_i$ values exceeded 100 pmC (which following Eq. 2, occurs when the DTW <2.67 mbg), which occurred more frequently in the Ovens/ Goulburn-Broken catchments (Fig. 3b). The application of Eq. 2 to determine $^{14}C_i$ led to three young water samples (out of 51) for the Limestone Coast where the uncorrected residence times were >1000 years (Fig. 3a). The greatest difference between the DTW-corrected and conventional MRTs for the Limestone Coast data was 9250 years, which corresponded to the $^{14}C$ measurement with the deepest DTW (68.8 mbg, Fig. 3c). On average, the MRTs were approximately 1500 years younger for the DTW-corrected MRTs, relative to the conventional MRTs for the Limestone Coast data (Fig. 3a, 3c).

The MRTs from the Ovens/ Goulburn-Broken catchments (Fig. 3b, d) were generally much younger than the Limestone Coast samples. For the Ovens/Goulburn-Broken catchments, the greatest difference between the DTW-corrected and conventional MRTs was 5410 years, with an average difference of approximately 300 years. As was the case with the Limestone Coast samples, the largest difference in estimated residence times was produced for the sample with the largest depth to water at the time of sampling (41.3 mbg). The average difference between the DTW-corrected and conventional MRTs was skewed for the Ovens/ Goulburn-Broken catchments by the fact that many of the calculated MRTs were young (<500 years) for both the conventional and DTW-corrected approaches. Five samples for the Ovens/Goulburn-Broken catchments were deemed to be young (<1000 years) after the application of the DTW-correction. Interestingly for the Ovens/Goulburn-Broken catchments, one sample was originally categorised as fossil water (>12,000 years) was no longer classified as fossil with the application of the DTW-correction approach.

The MRTs presented in Fig. 3 used 100 pmC as the $^{14}C_i$ value and best fit correction scheme to determine $^{14}C_i$ (Eq. 2). Fig. 4 includes the use of Eqs. 2-4 to determine $^{14}C_i$ values, thus providing lower and upper bounds on the estimated MRTs from the DTW correction approach. Tabulated values of the MRTs presented in Fig. 4 are available in Table S3.

{Approx. location of Fig. 4}

The application of Eqs. 3 and 4 to determine $^{14}C_i$ increases the range in depth-corrected MRTs for both the Limestone Coast and Ovens/ Goulburn-Broken catchments. For the Limestone Coast samples (Fig. 4a), two of the samples that were considered fossil water when $^{14}C_i$ was calculated from the best fit equation (Eq. 2), were not considered fossil water if Eq. 4 (lower bound, see Fig. 2) was used. For the Limestone Coast data, 18/51 samples were below the 500-year bounds placed on the minimum estimated MRT when Eq. 4 was used to determine $^{14}C_i$. The majority of MRT estimates for the Ovens/ Goulburn-Broken catchments were young water when using Eq. 2 to determine $^{14}C_i$ (Fig. 4b). Approximately 70% of the samples from the Ovens/ Goulburn-Broken catchments had estimated MRTs ranging between young water (500 y here) and ~3000 years.


### 3.3 Comparisons between $^{14}$C-based residence time and tritium activities

The Ovens/ Goulburn-Broken catchment $^{3}$H data (Table S2, Fig. 5) provides the opportunity to assess the $^{14}$C-based MRT estimates. Fig. 5 shows the raw $^{14}$C and $^{3}$H activities. Combined $^{3}$H and $^{14}$C$_{gw}$ data allow mixing to be identified (Le Gal La Salle et al., 2001; Favreau et al., 2002; Cartwright et al., 2007, 2017). The $^{3}$H peak

produced by the atmospheric nuclear tests in the southern hemisphere was signifcantly lower than in the northern hemisphere and $^{3}$H activities of groundwater recharged at that time are lower than those of modern rainfall (Morgenstern et al., 2010). Fig. 5 shows $^{3}$H and $^{14}$C$_{gw}$ activities in the Ovens and Goulburn-Broken groundwater (data from Table S2). Unsurprisingly, there is a general correlation between $^{14}$C$_{gw}$ and $^{3}$H. The shaded fields (Fig. 5) show the predicted covariation of these isotopes for the case where no macroscopic mixing between old and

young groundwater has occurred. This was constructed for a variety of flow geometries, the $^{3}$H record of rainfall in Melbourne, and the southern hemisphere atmospheric $^{14}$C$_{gw}$ record following Le Gal La Salle et al. (2001) and Cartwright et al. (2007, 2017). Samples lying outside these shaded regions most likely record the mixing between old and young groundwater. Closed system calcite dissolution lowers $^{14}$C activities (lighter region in Fig. 5 is for 15% calcite dissolution, i.e. $q = 0.85$). Samples lying to the right of the co-variance fields have relatively low $^{14}$C

activities but measurable $^{3}$H and would generally be interpreted as mixtures of young recently recharged water and older water flowing through the aquifer. However, because an additional reduction of $^{14}$C$_i$ may occur due to dilution of unsaturated zone $^{14}$CO$_2$, some of these samples may not show mixing.

{Approx. location of Fig. 5}

### 4. Discussion


The DTW-corrections (Eqs. 2-4) to estimate $^{14}$C$_i$ values proposed here are strongly influenced at depth (> 60 m) by the omission of the $^{14}$C$_{uz}$ data from the Yucca Mountain site (e.g., Kukler, 1969; Yang et al., 2015). The Yucca Mountain data were excluded from the data fitting process owing to complications induced by drilling at the site. Thorstenson et al. (1998) highlight that drilling in the 1980s provided conduits for modern atmospheric gases to

enter the system, thus providing explanations for the elevated $^{14}$C$_{uz}$ values at depths up to 400 m. The DTW-corrections to estimate $^{14}$C$_i$ presented here could be significantly revised, particularly at depths exceeding 60 m through the inclusion of further $^{14}$C$_{uz}$ data. The inclusion of the $^{14}$C$_{uz}$ data from Kukler (1969) and Yang et al. (1985) would lead to higher $^{14}$C$_i$ values determined from Eq. for depths exceeding 60 m (Eq. 2) and wider uncertainty bands (Eqs. 3, 4) over the same depth range. The exclusion of the Yucca Mountain data in the

generation of the DTW-correction relationships had only a minor influence on the interpretations of MRTs in the Limestone Coast and Ovens/ Goulburn-Broken catchments (Figs. 3, 4), owing to the depths to the water table at the time of sampling. Of the 113 $^{14}$C$_{gw}$ samples used to estimate MRTs, only two samples had water tables that were 30 m or more below the land surface. The fact that the Eqs. 2-4 rely on data from a single study for depths below ~60 mbg (e.g., Walvoord et al., 2005), is a limitation for the application of the DTW-correction approach

outlined here for sites with deep water tables.

The DTW-correction approach presented here is straightforward to apply as it requires only a measurement of the depth to the water table at the time of sampling and does not require additional data. The application of the DTW-



correction was illustrated here using conventional $^{14}C$ ages, but it could be easily incorporated into the input

function for lumped parameter models or numerical models. The DTW at the sampling well was used to estimate the $^{14}C_i$ values for the analyses in the demonstration of the method presented here. An alternative approach could be to estimate the DTW in the recharge zone, particularly in cases where there may be significant differences between the recharge zone and sampling location. For example, Wood et al. (2017) used the relationship between DTW and $^{14}C_{uz}$ for the Ti Tree Basin (central Australia) to generate spatially variable $^{14}C$ inputs in a regional scale

solute transport model.

Incorrect assumptions related to $^{14}C_{uz}$ can lead to significant over-estimation of MRTs (see Fig. 3). For example, the relationship presented in Fig. 6 shows the potential errors in MRT estimates from the assumption that $^{14}C_i =$ 100 pmC, where $^{14}C_i$ equals the value on the $x$-axis. To relate the error in MRT estimates to the depth to the water

table, the corresponding depths to the water table from Eq. 2 are presented on the secondary $x$-axis. Fig. 6 demonstrates that the assumption of $^{14}C_i = 100$ pmC for 24 m deep water table could lead to an over-estimation of MRTs on the order of 3000 years. Errors in estimated MRT can have major implications for recharge estimation. For example. Wood et al. (2015) highlighted that errors in MRTs caused by assumptions of initial $^{14}C$ activities can lead to errors in recharge estimates that exceed an order of magnitude. Furthermore, ignoring the

possibilities of low $^{14}C_{uz}$ values complicate the assessment of mixing using joint $^{14}C$ and $^3H$ data.

{Approx. location of Fig. 6}

## 5. Concluding remarks

Due to advances in technology, particularly the development of accelerator mass spectrometry in the case of $^{14}C$,

hydrogeologists can measure radioisotopes with higher throughput, lower detection limits, and higher precision than was the case 20 to 30 years ago. This has permitted more comprehensive studies to be carried out and led to a rapidly growing database of $^{14}C$ measurements. The continuing realisation, however, that there are numerous processes aside from radioactive decay that affect $^{14}C$ activities means that we are less certain in translating these data into useful parameters such as residence times. That some of these processes, such as those discussed here,

occur in the unsaturated zone, prior to recharge, adds to the complexity as these are difficult to resolve using groundwater data alone. Previous studies have focused on the importance of calculating $q$ in mean residence time estimates (e.g., Coetsiers and Walraevens, 2009; Han and Plummer, 2016). Here, the importance of accounting for processes in the unsaturated zone is demonstrated.

The analysis of $^{14}C$-depth relationships in the unsaturated zone is currently hampered by a relatively small number of studies, especially at the deeper levels. However, given that the impact on calculated mean residence times may exceed that caused by closed system calcite dissolution, further attention is warranted. Data from a wider range of environments would also help in assessing the variability of these relationships.





**Author contributions**

**DJI:** Conceptualisation, Data curation, Methodology, Investigation, Formal analysis, Writing-Original draft preparation, Visualisation, Supervision. **CW:** Conceptualisation, Data curation, Methodology, Writing-review and editing, Supervision. **IC:** Methodology, Formal analysis, Writing-review and editing. **TO:** Data curation, Formal analysis, Writing-review and editing.

**Competing interests**

The authors declare that they have no conflict of interest.

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





**Table 1: Summary of studies of $^{14}C_{uz}$. Sample depths are presented in metres below ground (mbg).**

| ID (Fig. 1) | Reference | Country, region | Sample depths (mbg) | Comments |
|---|---|---|---|---|
| 1 | Walvoord et al., (2005) | U.S.A., Amargosa Desert Research Site, Nevada | 0.2 to 106.8 | Data digitised and converted to pmC from Figure 4, Walvoord et al. (2005). |
| 2 | Yang et al., (1985) | U.S.A., Yucca Mountain, Nevada | 0.0 to 367.9 | Potassium hydroxide (KOH) based $^{14}C$ values used. Geology includes complex fractured tuff. |
| 3 | Thorstenson et al., (1998) | U.S.A., Yucca Mountain, Nevada | 0.3 to 10.1 | Study highlights advection of air through the unsaturated zone at Yucca Mountain site. |
| 4 | Bacon and Keller, (1998) | Canada, Saskatchewan | 0.2 to 7.1 | Samples influenced by organic matter. |
| 5 | Kukler, (1969) | U.S.A., New Mexico | 24.6 to 85.9 | Soil gasses collected in Bandelier Tuff. Atmospheric and biogenic sources of $^{14}C$ cited as plausible. |
| 6 | Haas et al., (1983) | U.S.A., North Dakota | 5.8 to 13.7 | Organic material (lignite) present. |
| 7 | Thorstenson et al., (1983) | U.S.A. North Dakota/ Texas | 5.0 to 44.5 | - |
| 8 | Reardon et al., (1980) | Canada, Ontario | 3.0 to 7.0 | Single measurement included in Thorstenson et al. (1983) |
| 9 | Fritz et al., (1978) | Canada, Ontario | 1.0 to 7.3 | - |
| 10 | Gillon et al., (2009) | France, Paris Basin | 1.2 to 4.5 | - |
| 11 | Gillon et al., (2009) | France, Herault region | 0.8 to 22.8 | - |
| 12 | Carmi et al., (2009) | Israel, Coastal aquifer of Israel | 2.5 to 13.5 | Organic material present. |
| 13 | Wood et al., (2014) | Australia, Ti Tree Basin, Northern Territory | 8.2 to 31.5 | - |
| 14 | Leaney and Allison, (1986) | Australia, Murray Basin, South Australia | 5.0 to 38.3 | Deep rooted vegetation present. |



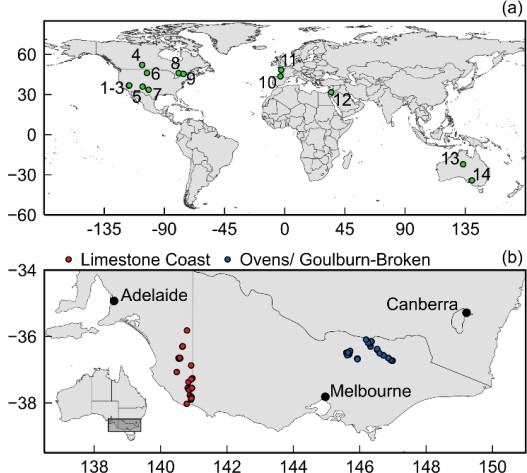

**Figure 1: Locations of $^{14}$C data used. (a) Locations of studies of unsaturated zone $^{14}$C ($^{14}C_{uz}$), where 1 = Walvoord et**
**al., (2005), 2 = Yang et al., (1985), 3 = Thorstenson et al., (1998), 4 = Bacon and Keller, (1998), 5 = Kukler, (1969), 6 =**
**Haas et al., (1983), 7 = Thorstenson et al., (1983), 8 = Reardon et al., (1980), 9 = Fritz et al., (1978), 10 and 11 = Gillon**
**et al., (2009), 12 = Carmi et al., (2009), 13 = Wood et al., (2014), 14 = Leaney and Allison, (1986). (b) Locations of**
**groundwater $^{14}$C ($^{14}C_{gw}$) data. Red markers show locations $^{14}C_{gw}$ samples from the Limestone Coast region of South**
**Australia. Blue markers show locations of $^{14}C_{gw}$ samples from the Ovens/ Goulburn-Broken catchments, Victoria.**


**Figure 2: Relationship between pmC in the unsaturated zone ($^{14}C_{uz}$) and depth below the surface for sedimentary**
**basins. Data points omitted from the fitting process (grey crosses) were omitted due to the presence of high organic**
**matter, deep rooted vegetation, or where gasses were influenced by the advection of modern air into the unsaturated**
**zone.**



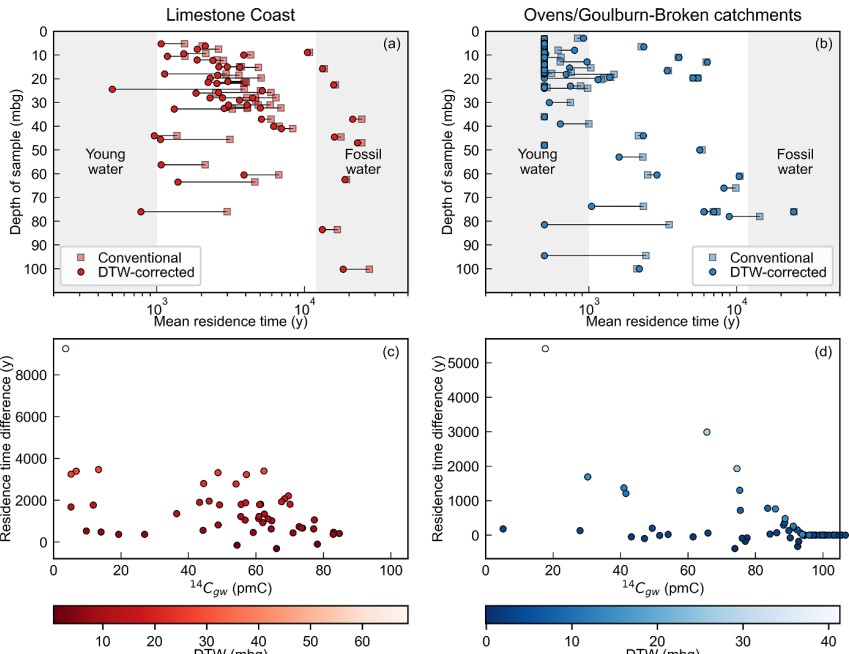

**Figure 3: Calculated MRTs for the Limestone Coast (left column, red colours) and the Ovens/ Goulburn-Broken catchments (right column, blue colours). Panels (a) and (b) show the difference in calculated MRTs for the uncorrected**
**(Eq. 1 where $^{14}C_i$ = 100 pmC, square markers) and the depth corrected $^{14}C_i$ values (circle markers). Panels (a) and (b) use definitions of young water of ≤1000 years (e.g., Cartwright et al., 2020), and fossil water of ≥12,000 years (Jasechko et al., 2017). Panels (c) and (d) show the difference in MRTs between the use of 100 pmC and the value calculated in Eq. 2 against the measured $^{14}C$ activity in groundwater ($^{14}C_{gw}$). Marker colours in (c) and (d) are based on the depth to the water table (mbg) when the sample was collected.**



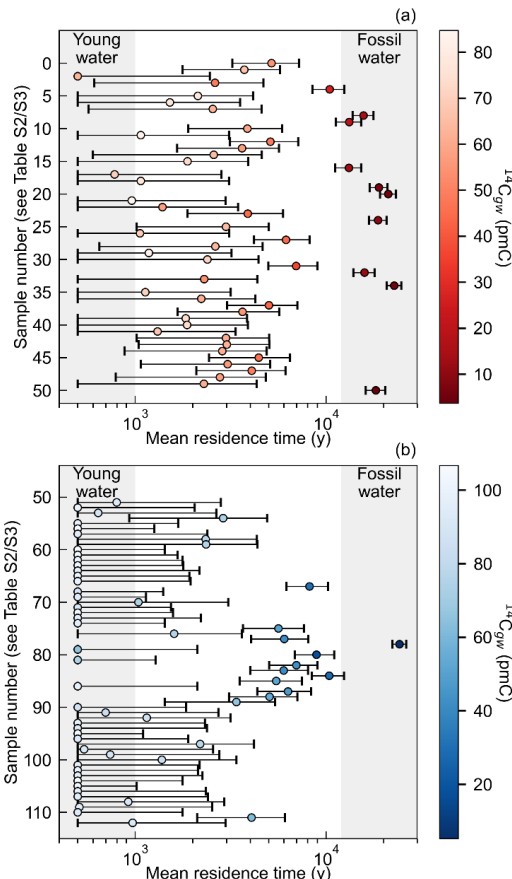


**Figure 4: Mean residence times (y) determined using the best fit relationship (Eq. 2, circle markers), the upper limit (Eq. 3, upper whisker) and the lower limit (Eq. 4, lower whisker). For sample numbers (y-axis), see Table S2 and S3 in the supporting information. (a) MRTs from the Limestone Coast region, (b) MRTs from the Ovens/ Goulburn-Broken catchments. In both panels, shading denotes the measured $^{14}C_{gw}$ activities.**


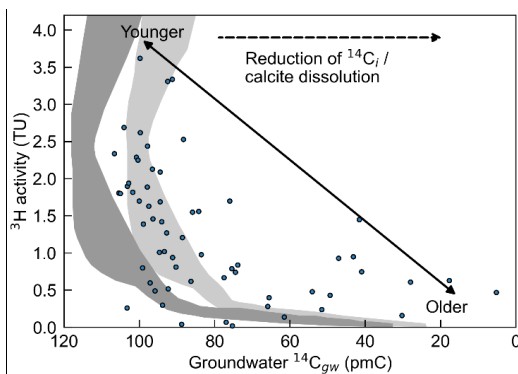

**Figure 5: Comparisons between measured $^3$H and $^{14}C_{gw}$ activities and MRTs for the Ovens/ Goulburn-Broken catchments (see Fig. 2). Note, the x-axis ($^{14}C_{gw}$) has been inverted. Grey shading denotes the predicted covariation of $^3$H and $^{14}C$ for the case where no macroscopic mixing between old and young groundwater has occurred, dark grey**
**for q = 1, light grey for q = 0.85.**



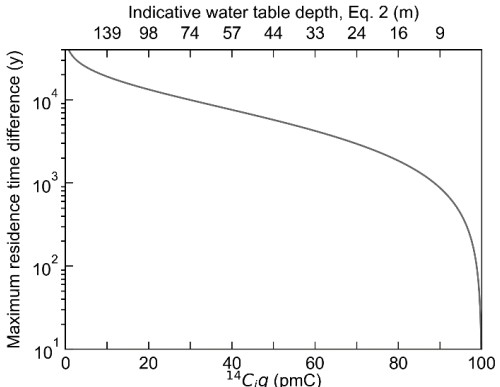

**Figure 6: Maximum difference in calculated MRT (y) where $A_0$ is assumed to be 100 pmC and the MRT calculated using the $A_0$ value on the *x*-axis.**