# Peer review of "Depth to water table correction for initial carbon-14 activities in groundwater mean residence time estimation"

_Hydrology and Earth System Sciences, 2021_

## Author Response (AR1)

Thank you for the opportunity to revise our manuscript (minor revisions). We have modified the original response to referee document below to describe changes made to the manuscript. All mention of line numbers relate to the track changes version of the document.

**Anonymous Referee #2**

General comments

This paper uses literature values of unsaturated zone $^{14}$C activities to develop a depth to water (DTW) correction for initial $^{14}$C values for groundwater dating. Not all previous studies have assumed that the unsaturated zone is in equilibrium with the atmosphere, but many have. In these cases, the correction indicated by the equation can be substantial (corrected mean residence times (MRTs) can be thousands of years younger than the uncorrected MRTs). These effects are well known, as attested by the 14 studies used to develop the DTW correction, but the contribution here is the development of the correction equation, which will be easy and useful for others to adopt.

Response: Thank you for the comments. Yes, our goal was to bring the findings from the unsaturated zones together and generalise them, thereby presenting simple to apply approaches to account for this specific process in the unsaturated zone in the estimation of mean residence times of groundwater.

We address your additional comments below.

Logically, the DTW in the recharge area has the most relevance to the correction required, not the DTW where the sample was collected. Using the DTW from the sample location (as in this paper) is a compromise made for convenience.

Response: As the referee highlights, the selection of water levels in the wells was made for convenience. We note that Marina Gillon raised this point in her comments also.

As the focus of the manuscript is on the presentation of the method (with a demonstration), we feel that this approach is appropriate. To highlight that the DTW in the recharge zone may be of more importance, we have added the following text to the end of Section 2.2 (Saturated zone data collation) (lines 103 to 106):

It is likely that the DTW in the recharge zone is more relevant. One approach could have been to determine DTW from spatially mapped water levels (e.g. Wood et al., 2017). Nonetheless, the simple approach to estimate DTW from the sampled wells allows for a demonstration of the methods outlined here.

We further discuss the implications of using DTW from a sample well, relative to the recharge zone (where the water table is expected to be closer to the surface) on lines 286-293:

The example applications presented here used the DTW at the sampling well was used to estimate the 14Ci values. These DTW values are likely greater than the DTW at the recharge zone, at the time of recharge, leading to minor over-corrections of 14Ci values from Eqs. 2-4. For example, for Well ID 7022-128 (Sample ID = 16, Clgw = 13.35 pmC, DTW = 27.47 m, see Table S3) the MRT using Eq. 2 was 13,180 y. If the DTW was assumed to be 5 m shallower (22.47 m), the MRT increased to 13,880 y (700 y, or ~5%). Given that Eqs. 2-4 are straightforward to implement, the impact of uncertainty on the DTW could be easily investigated.

The paper makes the implicit assumption that only the residence time in the saturated zone is of interest. Time spent passing through the unsaturated zone in the recharge zone presumably is assumed to be negligible or of no interest (which of these is not specified as this issue is not mentioned in the paper).

> Response: It is true that groundwater recharge is not instantaneous. However, the timescales of infiltration through the unsaturated zone is likely to be a few weeks to a few years, which is short relative to the several thousand-year time frame of 14C residence times that are typical of many aquifers. This point is made on lines 152-154:
>
> The calculation of MRTs ignores the time taken for water to infiltrate through the unsaturated zone. The timescales of infiltration are expected to be on the order of a few weeks to a few years, which is short relative to the several thousand-year timeframes of 14C-based MRTs.

The authors have adopted a very simplified MRT estimation procedure, which they label "conventional". It is hardly conventional, since it ignores (1) the recent history of $^{14}$C activity in the atmosphere due to nuclear weapons testing (instead they assume a uniform atmospheric activity), (2) the input of $^{14}$C-free carbon from the aquifer matrix (i.e. they assume q = 1), and (3) groundwater dispersion producing a distribution of residence times in the sample (in effect assuming piston flow). I think it could be described better as "simplified". However, as an exercise to illustrate the application of the correction equation it is reasonable.

> Response: The use of the term 'conventional' to describe the assumptions highlighted by the referee is commonplace in hydrogeology. To address this comment, the phrase "so-called" has been inserted into the first description of conventional ages and add an additional reference clarify our use of 'conventional' in this context. The relevant sentence in paragraph two of the introduction (lines 41-43) now read:
>
> This approach yields so-called conventional radiocarbon ages in years Before Present (BP) where 1950 AD = 0 years BP (Clark and Fritz. 1997; Plummer and Glynn, 2013; Cartwright et al., 2020).

The paper is well organised and succinct, but possibly too succinct in parts making it unnecessarily difficult to understand. (e.g. The caption of Fig. 6 is very unhelpful. The symbol $A_0$ from the caption is not used in the text.)

> Response: A0 was replaced with Ciq (to be consistent with the text). To ensure that the purpose of Figure 6 is clear, the caption for Fig. 6 was revised to:
>
> Figure 6: Maximum difference in calculated MRT (y) where Ciq on the x-axis is used, relative to the case where it is assumed to be 100 pmC. Secondary x-axis shows indicative water depths that correspond to 14Ciq values shown on the lower x-axis according to Eq. 2.

However, the paper is generally clearly written with few technical or detail corrections needed. It is suitable for the journal and has no unnecessary or overlong sections. The references are appropriate. The data set is sufficient to support the discussion and conclusions. Title and abstract are satisfactory. I think the paper should be published after minor revision.

Response: We thank the reviewer for their comments on our manuscript.

Specific comments

L124-125. Not sure what this sentence means. "However, owing to the relatively small sample size, the data was included in the fitting process independent of the year in which it was collected." Does this mean that no account was taken of the actual $^{14}C$ input function?

Response: The reviewer is correct. The paragraph in question identifies difficulties in accurately estimating what the actual $^{14}C$ input function might be (i.e. it will differ from atmospheric concentrations). Thus, our fitting process did not account for the year that the sample was collected in. The sentence now reads as:

Line 128-130: Owing to the abovementioned complexities, the sample date was not taken into account in the fitting process.

L134-139. I would like to see the simplifying assumptions in itemised form (1, 2, 3)

Response: Changes made. The simplifying assumptions are itemised in the manuscript. See paragraph from line 140.

**Review by Marina Gillon**

General comments
The authors propose a simple method to determine the initial carbon-14 activity of groundwater for $^{14}C$-dating of groundwater, using an empirical relationship between the depth and $A^{14}C$ of $CO_2$ in the unsaturated zone. This approach is very interesting as the role of the unsaturated zone can be taken into account even when data of the unsaturated zone are not available.

Response: Thank you. We expect that the simple equations provided here will be useful to account for the unsaturated zone, even when data are not available.

I have four main remarks and questions: (numbers, i.e. (1), below added by manuscript authors)
(1) The relationship between depth and $A^{14}C$ of $CO_2$ in the unsaturated zone have been determined from data measured after nuclear tests. Is it reasonable to use this approach for water recharged before nuclear tests?

Response:
Prior to nuclear testing, $^{14}C$ activities in the atmosphere were ~100 pmC. Now, in the post-peak times, atmospheric activities are on the order of ~105 pmC (e.g. Hua et al., 2013). The analyses can be performed, but as with the case of all approaches, users of the techniques should take care with interpreting results.

The following has been added to highlight the point above (lines 194-195):

As with the application of other correction schemes, care should be taken in interpreting MRTs determined using of Eqns. 2-4 to determine 14Ci values.

(2) The evolution of $A^{14}C$ of $CO_2$ in the unsaturated zone is in part linked to gas-water-rock interactions (and organic matter for some sites). These interactions modify both $A^{14}C$ and $\delta^{13}C$ of $CO_2$. Several correction models in carbonated aquifers use the $\delta^{13}C$ of soil $CO_2$. Isn't

there a risk of over-correction of the effect of water-carbonate interactions if the $A^{14}C_i$ is already modified in the unsaturated zone and no the $\delta^{13}C$? Use the $\delta^{13}C$ of $CO_2$ for the groundwater level depth (as for $A^{14}C$) would probably avoid this problem, which means that a relation between depth and $\delta^{13}C$ of $CO_2$ would also necessary.

> Response: The approach that we discuss here is independent of the subsequent corrections for addition of 14C-free C from the aquifers that is typically due to closed-system calcite dissolution. In many cases, the use of 13C to calculate the extent of dilution with 14C-free carbon is problematic (due to: the calcite 13C being poorly constrained; uncertainties in the 13C of recharge; and other processes such as open-system calcite dissolution, methanogenesis, and/or input of geogenic CO2). In some cases, those processes lead to over-corrected 14C residence times. We highlight the fact that profiles of d13C with depth have been shown to be near-vertical (lines 130-133).

> Additionally, while many of the unsaturated zone studies also included δ13C measurements, these data were not utilised here as δ13C-depth profiles have been shown to be almost vertical with depth (e.g., Walvoord et al., 2005; Wood et al., 2017), containing little additional information

(3) The depth of groundwater level used in the calculations is important due to the depth of water level in the borehole where water is collected is not necessary the same of groundwater level in recharge area (especially for confined aquifers) and varies in time. Authors talk rapidly of this problem, in the last part of paper. Perhaps, authors should talk about it earlier in the text, and justify their choice of groundwater level for their sites (at the sampling borehole and no in recharge area). They should also discuss about uncertainties associated to the choice of groundwater level (recharge area or sampling location; time variation), does these uncertainties be problematic or negligible?

> Response: We agree that this is an important point that was also raised by the anonymous referee. We address this comment by extending the discussion on this point earlier in the manuscript, and in the discussion. The two sections of text now read as:

> Text to add to Section 2.2 (lines 103-106) - Saturated zone data collection:
> It is likely that the DTW in the recharge zone is more relevant. One approach could have been to determine DTW from spatially mapped water levels (e.g. Wood et al., 2017). Nonetheless, the simple approach to estimate DTW from the sampled wells allows for a demonstration of the methods outlined here.

> Text to add to Section 4 – Discussion (lines 286-293):
> The example applications presented here used the DTW at the sampling well was used to estimate the 14Ci values. These DTW values are likely greater than the DTW at the recharge zone, at the time of recharge, leading to minor over-corrections of 14Ci values from Eqs. 2-4. For example, for Well ID 7022-128 (Sample ID = 16, Clgw = 13.35 pmC, DTW = 27.47 m, see Table S3) the MRT using Eq. 2 was 13,180 y. If the DTW was assumed to be 5 m shallower (22.47 m), the MRT increased to 13,880 y (700 y, or ~5%). Given that Eqs. 2-4 are straightforward to implement, the impact of uncertainty on the DTW could be easily investigated.

(4) The geology of sites where the $A^{14}C$ of $CO_2$ have been measured is not indicated in the paper. It is important to indicate and discuss it because the gap between min and max relationship between depth and $A^{14}C$ of $CO_2$ in the unsaturated zone can be a consequence

of differences in geologic properties of aquifers (porous aquifer, fractured aquifer, presence or not of carbonate minerals…).

> Response: An additional column "Geological description of site" has been added to Table 1 to provide descriptions of geology from unsaturated zone $^{14}$C sites.

Specific comments
 L96-103 see general comment N°3

> Response: As per comment (3) above, we provided additional discussion to justify this reason why this decision was made, and its potential implications. See comment addressing comment (3) for details.

L105-109 More details about the method or a reference where details are given, would be interesting.

> Response: Additional explanation that the fitting approach has been added. The approach used determined two unknowns a and b in the equation Cluz = a exp(bz). The updated explanation is now on lines 110-112:
>
> The unsaturated zone sample depth-14Cuz relationship was produced by fitting the 14Cuz and sample depth data (Table S1) using the curve_fit function in the scipy.optimize library and the nominal_values function from the uncertainties.unumpy libraries in Python. The curve fitting approach was used to determine the coefficients a and b in the equation Cluz = a exp(bz). This approach also was used to find the best fit to the data, as well as to produce upper and lower bounds on the best fit relationship based on the standard deviation of the observed data.

L124-125 I don't understand the link between small size of sample and the fact to not take into account the sampling year. Year-to-year variability can exist regardless of the sample size.

> Response: We agree that the sentence appears unclear. The goal of the paragraph was to highlight that accounting for the complex input function to groundwater (related to, but not the same as in the atmosphere) is challenging. An option could have been to produce multiple lines of best fit for time periods, for example (i.e. each line would be informed by less data).
>
> We rephrased the sentence to focus on the challenge of incorporating the complex input function. This change was made on lines 128-130:
>
> Owing to the abovementioned complexities, the sample date was not taken into account in the fitting process.

L216-225 you should also compare the results of min or max relationship with the calculation using the A$^{14}$C$_i$ equal to 100pMC and discuss it.

> Response: To ensure that figures were legible, the approach taken in the original submission was to (1) show corrected (mean, i.e. Eq. 2 to determine 14Ci) vs. uncorrected mean residence times (Figure 3), and then to show the min (Eq. 4), mean (Eq. 2) and max (Eq. 2) results. Our preference is to retain the plots as presented. No change made.

L235-236 More information could be provided on the construction of the envelopes on the figure 5, especially what do you mean by « variety of flow geometry » ?

Response: The envelopes in Fig. 5 were constructed following Cartwright (2017; doi: 10.1016/j.jhydrol.2017.10.053) using a range of lumped parameter models that relate the 14C and 3H activities of groundwater with different mean residence times to the input function of these tracers. The input functions are those appropriate to southeast Australia and the exponential piston flow and dispersion models were used to create the envelopes. Text has been added to highlight this point on lines 249-250:

(i.e. using the exponential piston flow, and dispersion models, see Cartwright et al., 2017),

L239-242 (and fig 5): Is it possible to differentiate the samples lying to the right due to a mixing between young and old water and the samples lying to the right due to an $A^{14}C_i$ different from 100pMC?

Response: That would be the case if a uniform A0 of 100 pMC were used. However, the curves on Fig. 5 were constructed using an input function based on the variation of 14C in the atmosphere (McCormac et al, 2004m, doi: https://doi.org/10.1017/S0033822200033014.). Samples lying to the RHS of the curves have probably undergone mixing. Samples to the LHS would be over-corrected (i.e. their A14C is too high: e.g. Cartwright et al., 2013, doi: 10.1016/j.apgeochem.2012.10.023).

The comment on data that lie to the right was already included in the manuscript. Observations on the data that lie to the left was added, with a reference to a paper by Cartwright et al. 2013 for readers who wish to read further. New text reads (on line 259-260):

Samples to the left are likely over-corrected, whereby 14Ci values are too high (e.g., see Cartwright et al., 2013).

L265-266 see general comments N°3

Response: The point of the sentence identified was to highlight that the excluded data from the Yucca site was generally very deep. The water levels in the wells used in our manuscript was generally very shallow (and the water levels in the recharge zone would be shallower still). We clarify this point in the revised manuscript, lines 270-274:

The exclusion of the (generally deep) Yucca Mountain data in the generation of the DTW-correction relationships had only a minor influence on the interpretations of MRTs in the Limestone Coast and Ovens/ Goulburn-Broken catchments (Figs. 3, 4), owing to the depths to the water table at the time of sampling. This observation would hold even in the case where DTW values from the recharge zone, rather than sampling wells were used.

Caption of figure 6 : You talk about $A_0$ whereas you use $A_i$ in the text. Does $A_0$ correspond to $qA_i$ ? Why do you not use $A_i$ in the figure 6 in order to show only the role of the unsaturated zone ? Have the depths indicated on the second x-axis been calculated for q = 1? it should be specified.

Response: Yes, A0 referred to $^{14}Ci_q$ (i.e. qAi using your notation). Caption now reads as:

Figure 6: Maximum difference in calculated MRT (y) where 14Ciq on the x-axis is used, relative to the case where it is assumed to be 100 pmC. Secondary x-axis shows indicative water depths that correspond to 14Ciq values shown on the lower x-axis according to Eq. 2. Result assumes q = 1

We would like to thank Dr Gillon for her constructive review.